# A Scoping Review of Landform Classification Using Geospatial Methods

**Zama Eric Mashimbye * and Kyle Loggenberg**

Department of Geography and Environmental Studies, Stellenbosch University, Stellenbosch 7602, South Africa
* Correspondence: erim@sun.ac.za; Tel.: +27-218-089-702

**Abstract:** Landform classification is crucial for a host of applications that include geomorphological, soil mapping, radiative and gravity-controlled processes. Due to the complexity and rapid developments in the field of landform delineation, this study provides a scoping review to identify trends in the field. The review is premised on the PRISMA standard and is aimed to respond to the research questions pertaining to the global distribution of landform studies, methods used, datasets, analysis units and validation techniques. The articles were screened based on relevance and subject matter of which a total of 59 articles were selected for a full review. The parameters relating to where studies were conducted, datasets, methods of analysis, units of analysis, scale and validation approaches were collated and summarized. The study found that studies were predominantly conducted in Europe, South and East Asia and North America. Not many studies were found that were conducted in South America and the African region. The review revealed that locally sourced, very high-resolution digital elevation model ( DEM) products were becoming more readily available and employed for landform classification research. Of the globally available DEM sources, the SRTM still remains the most commonly used dataset in the field. Most landform delineation studies are based on expert knowledge. While object-based analysis is gaining momentum recently, pixel-based analysis is common and is also growing. Whereas validation techniques appeared to be mainly based on expert knowledge, most studies did not report on validation techniques. These results suggest that a systematic review of landform delineation may be necessary. Other aspects that may require investigation include a comparison of different DEMs for landform delineation, exploring more object-based studies, probing the value of quantitative validation approaches and data-driven analysis methods.

**Keywords:** landform classification; geomorphometry; land surface; terrain analysis; topography; DEM

## 1. Introduction

This study presents a scoping review on landform delineation based on geospatial techniques. Scoping reviews have recently become common for evidence synthesis. According to Munn et al. [1], scoping reviews are a valid approach in situations where systematic reviews are unable to meet the necessary objectives of knowledge users. While scoping reviews follow a similar and robust approach as systematic reviews, they perform different purposes. A variety of reasons for conducting scoping reviews have been advanced by different authors. Arksey and O'Malley [2] contended that a scoping review can be conducted to: examine the extent, range and nature of research activity; determine the value of undertaking a full systematic review; summarize and disseminate research findings; and identify research gaps in the existing literature. Munn et al. [1] expanded motivations for scoping reviews to incorporate: identifying the types of available evidence in each field; clarifying key concepts (definitions) in the literature; examining how research

is conducted on certain topics or fields; and identifying key characteristics or factors related to a concept. Conducting a scoping review could be beneficial for landform classification due to the complex nature of the field and the diversity of methods and applications, including rapid advancements in methods, tools and technology.

According to Xiong et al. [3], classifying landforms is beneficial for geomorphological applications (geomorphological, geological and ecological processes), gravity-controlled processes (runoff, erosion and mass movements) and radiative applications (viewshed and visibility analysis applications). Due to their differences relating to the physical processes that aided their formation and the difference in the way they function, it is essential to classify landforms (MacMillan & Shary [4]; Etzelmuller and Sulebak [5]). Landform classification entails categorizing terrain into areas of uniform characteristics relating to slope, soil, biological and physical processes. Mokarram and Sathyamoorthy [6] recorded that landform classification is a science of land definition that entails the extraction of land-surface parameters and objects from digital elevation models (DEMs) and digital terrain models (DTMs). MacMillan and Shary [7] defined a landform as a physical feature of the Earth's surface that possesses a characteristic, recognizable shape and is produced by natural causes. MacMillan and Shary [7] asserted that landform entities also differ with regard to shape, size, orientation, relief and contextual position. The philosophies and techniques of landform identification and categorization have also contributed to studying landscapes of other planets. For example, Bue and Stepinski [8] investigated a numerical method for classifying and characterizing landforms on Mars. They used unsupervised classification, based on a self-organizing map technique to divide all pixels using topographic attributes computed from a DEM into mutually exclusive and exhaustive landform classes based on the similarity between attribute vectors. They depicted the results as a thematic map of landforms and the accompanying attribute statistics, which were used to assign semantic meaning to the classes.

Classifying landforms can be based on manual and automatic methods. According to MacMillan and Shary [7], automated extraction of landforms is usually premised on replicating proposed manual classification schemes, of which there are several. These include schemes by Fenneman (1938), Veatch (1935) and Hammond (1954, 1964) for the USA, the Australian classification system of Speight (1974), Speight (1990), the SOTER4 Global Soil and Terrain Database (van Engelen and Ting-tiang, 1995), the ITC system of geomorphic mapping (Meijerink, 1988) and the geo-pedological approach by Zink (Hengl and Rossiter, 2003) [7]. Mokarram and Sathymoorthy [6] and Xiong et al. [3] recorded that classification approaches can be based on general geomorphometry or specific geomorphometry. General geomorphometry considers the detection of landforms as continuous features, whereas specific geomorphometry detects landforms as discrete features, for example, drumlins, sand dunes, alluvial fans and landslides. For instance, Brigham and Crider [9] investigated a methodology to probe the degradation of fault scarps in jointed bedrock by making field observations of seven fault carps in Hawaii, California and Iceland. They collected aerial imagery for Structure-from-Motion (SfM) photogrammetry. They utilized expert knowledge of a geomorphologist to manually classify fault-scarps profiles from SfM-derived point clouds into six morphologic categories and then used principal component analysis to quantitatively distinguish morphologic classes. Thereafter, they used supervised classification based on a support vector machine (SVM) to classify morphologic classes using the principal-component analysis coordinates of the classified profiles. They determined that drivers of scarp form could be quantitatively determined by analyzing the covariance between morphologic variability metric and other geomorphic parameters. Li et al. [10] used deep learning (DL) and random forest (RF) algorithms to automatically classify complex and transitional landforms using imagery, DEMs and terrain derivatives. They found that DL could classify transitional landforms better than RF.

Recent developments that include among others wide availability of digital data (DEMs and satellite imagery), analysis frameworks, analysis methods, analysis platforms

and tools are impacting the way terrain analysis and geomorphometry is conducted. Recent reviews by Mokarram and Sathymoorthy [6], Maxwell and Shobe [11], and Xiong et al. [3] uncovered most of these recent developments. Some of the key issues pertaining to DEM data that affect terrain analysis have been highlighted by Xiong et al. [3]. These include issues related to sources of data, DEM voids, DEM accuracies and security of using high-resolution DEMs [3]. For example, Verhagen and Dragut [12] investigated the value of object-based image analysis to automatically delineate landforms using DEMs to predict and interpret the location of archaeological sites. They concluded that OBIA was suitable for the automatic delineation of landforms but requires an improved conceptual framework that is adapted to the local area and archaeological questions to improve the delineation and interpretation of the landforms. Dobre, Kovacs and Bugya [13] probed the utility and limitations of various open-source DEMs at various spatial resolutions for extracting geomorphic surface remnants in a semi-arid mountainous environment using a well-known open-source GRASS GIS Geomorphons module. They scrutinized peaks as remnants of geomorphic surfaces. They determined that irrespective of the characteristic differences in the accuracies of the DEMs used, all DEMs used were able to detect surface remnants appropriately. In Mokarram and Sathyamoorty [6], issues relating to definitions and attributes (for example morphometry, geomorphometric context, terrain positions and scale) of landforms and methods for classifying landforms are discussed. This study exposed issues relating to less adoption of landform categorization based on general geomorphometry in comparison to specific geomorphometry. In addition, trends regarding automated classification and fuzzy versus crisp classification were exposed. Gioia et al. [14] evaluated the accuracy of automatic landform classification on a large sector of the Ionian coast of southern Italy. They performed automatic landform classification using an algorithm based on the individuation of basic landform classes called geomorphons. They reported that automatic landform classification using the geomorphon-based method could achieve accuracies greater than 70% when compared to landforms mapped using traditional geomorphological analysis.

Against the backdrop of the recent wide availability of DEM and satellite imagery data and developments in analysis methods, platforms and tools, it will be beneficial to synthesize the state of knowledge of landform classification. To the best of the knowledge of the authors, no scoping or systematic review is available on landform classification. While a scoping review and systematic review follow a similar methodology to synthesize the state of knowledge, a scoping review is not as strict in terms of bias in the selection of articles and the number of articles used. Yet, it is very useful to identify crucial knowledge relating to assessing the potential for a systematic review, identifying potential gaps and uncovering how research is conducted for specific fields. This study aims to conduct a scoping review to map the state of knowledge about landform classification. The distribution of studies, datasets used, methods and validation approaches will be investigated. The results of this study will aid in identifying research gaps, revealing the state of knowledge on landform classification and identifying common methods and their implementation and applications.

## 2. Research Strategy

The scoping review was conducted following the reporting standards outlined by the Preferred Reporting Items for Systematic reviews and Meta-Analyses, better known as PRISMA [15]. The PRISMA reporting standards provide guidelines that reduce data collection biases and increase procedural objectivity. In so doing, PRISMA ensures analytical reproducibility and transparency of review protocols and has, therefore, been employed for various systematic and scoping reviews [16–18]. Additionally, this scoping review adopted the methodology proposed by Arksey and O'Malley [2], which comprises the following key elements: identifying the research questions; identification of relevant studies; selection of studies; charting the data; and collating, summarizing and reporting the results. Delineation of research questions, identification and screening of studies, and

extraction of relevant parameters are presented in the succeeding subsections. The information is then collated, summarized and presented in the results and discussion section. Concluding remarks about the results of the scoping review are then provided.

### 2.1. Research Questions

Four research questions were formulated to guide this scoping review. The research questions aim to highlight the knowledge gaps in recent literature and direct the mapping of various methodological and geographical trends in the research field. Table 1 presents the formulated questions.

**Table 1.** Research questions.

| Number | Research Questions (RQ) | Motivation |
|--------|--------------------------|------------|
| RQ1 | Where has landform classification been conducted? | To identify the geographical distribution of landform classification studies. |
| RQ2 | What datasets have been used to classify landforms? | To determine the data inputs commonly used to classify landforms. |
| RQ3 | How are landforms classified? | To identify trends and gaps in data analysis and application, including application scale. |
| RQ4 | How are landform classification analyses validated? | To understand how landform classification accuracy is assessed. |

### 2.2. Article Screening

The literature searches were constrained to research databases that were readily accessible to the authors, namely the EBSCOhost databases, SAGE Journals, ScienceDirect, Scopus, Taylor & Francis and Wiley Online Library. Articles were obtained based on the following search string:

*""Landform classification" AND (topography OR "land surface") AND (terrain OR morphometry OR geomorphometric)"*

The search string was consistently applied in each of the five research databases, producing a total of 566 articles. As the review aimed to focus primarily on recent developments in landform classification, the literature search was restricted from 2012 to 2022. After the automated removal of ineligible and duplicated articles, facilitated through the Mendeley Desktop Software, 306 articles remained. These articles were screened based on titles, keywords and abstracts. Once the initial screening was completed, a detailed review of 159 articles was performed to determine the final article selection. Full articles were assessed based on their accessibility, relevance and subject matter. A total of 59 articles were deemed eligible for this scoping review (see Table A1, Appendix A). A summary of the article screening and selection process is presented in Figure 1. The summary was produced according to the PRISMA guidelines using the R Shiny App developed by [19].

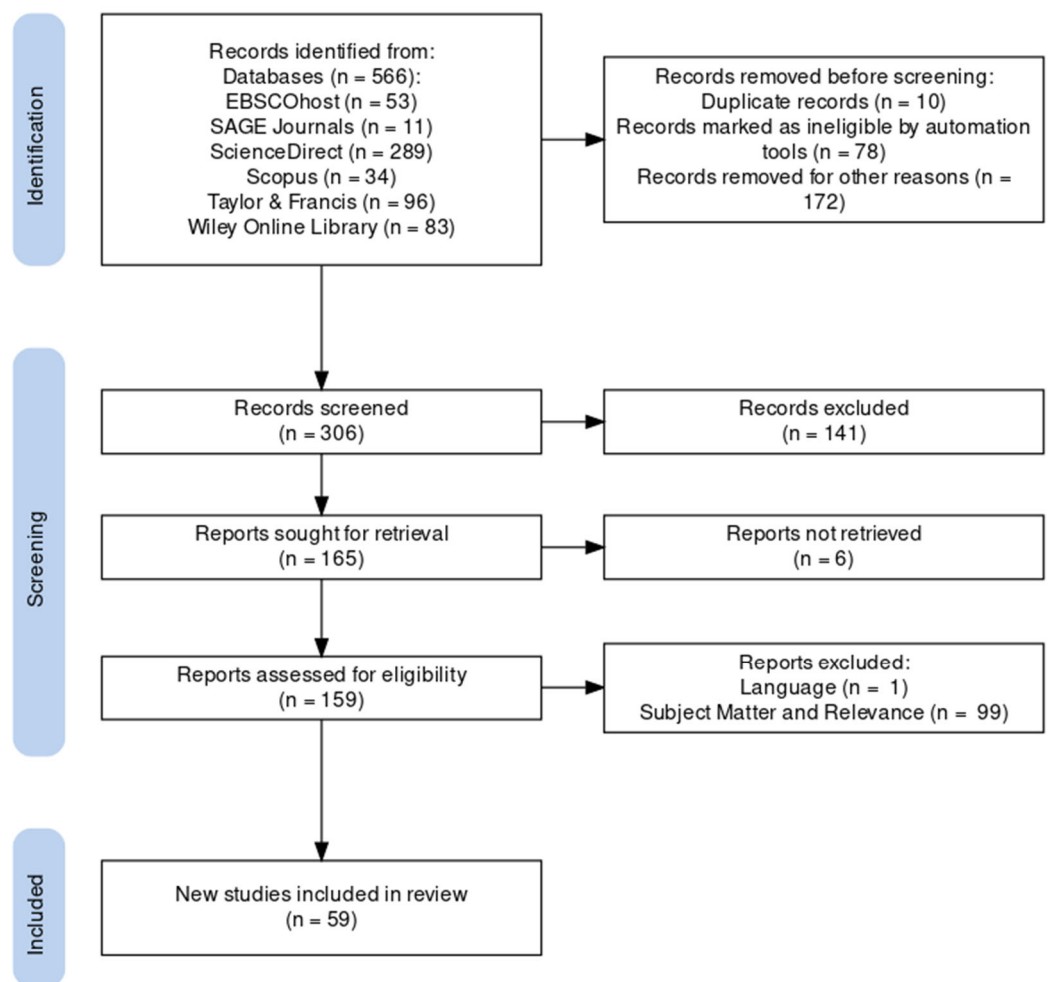

**Figure 1.** PRISMA article screening workflow.

*2.3. Data Extraction*

The final 59 articles were collaboratively reviewed by the two authors. The authors extracted information pertaining to the geographical distribution of the studies, datasets used, the scale at which the studies were conducted, methodological approaches and validation schemes. A coding process was followed, which was informed by the literature, to address RQ2–4. Table 2 outlines the coding scheme used in the review.

**Table 2.** Coding scheme used for data extraction.

| Research Questions | Category | Explanation | Source |
|---|---|---|---|
| RQ2 Datasets Used | DEM < 30 m | Digital elevation datasets with a spatial resolution less than 30 m. Considered as very high-resolution DEM data. | |
| | DEM = 30 m | Digital elevation datasets with a spatial resolution of 30 m. Considered as high-resolution DEM data. | |
| | DEM > 30 m | Digital elevation datasets with a spatial resolution greater than 30 m. Considered as moderate to coarse resolution DEM data. | |
| | Laser | Any elevation dataset derived from active laser sensors, including airborne Light Detection and Ranging (LiDAR) and Terrestrial laser scanners. | |

| | | | |
|---|---|---|---|
| | Digital Imagery | Any panchromatic, multispectral or hyperspectral datasets captured using digital sensors. | |
| | GPS measurements | Elevation datasets derived from in-field Global Positioning System (GPS) measurements. | |
| | Analogue | Any hardcopy maps or imagery (including any images developed using film photography). | |
| RQ2 DEM Source | ALOS DEM | Elevation data obtained from the **A**dvanced **L**and **O**bserving **S**atellite (ALOS). | |
| | ASTER GDEM | Elevation data obtained from the Terra **A**dvanced **S**paceborne **T**hermal **E**mission and **R**eflection Radiometer (ASTER). | |
| | Other | Any elevation datasets specifically created for the study area or sourced from local or national government authorities or private companies. Includes datasets created using data such as laser scanning, GPS measurements, etc. | |
| | STRM DEM | Elevation data obtained from the **S**huttle **R**adar **T**opography **M**ission (SRTM). | |
| | TanDEM-X | Elevation data obtained from the TanDEM-X interferometric Synthetic Aperture Radar (SAR) satellite mission. | |
| | USGS | Any elevation datasets obtained from the U.S. Geological Survey (USGS), including the Global Multi-resolution Terrain Elevation Dataset (GMTED). | |
| RQ3 Application Scale | Local | Studies conducted on a relatively small area such as a town, city, village, ward or suburb. | |
| | Regional | Studies conducted at a municipal, district, state or provincial level. | |
| | National | Studies conducted across the vast majority of a country's extent. | |
| | Global | Studies conducted across international borders and comprising the majority of the Earth's landmass. | |
| RQ3 Applications | Gravity-controlled processes | Studies relating to gravity-controlled processes, e.g., runoff, erosion and mass movements. Includes disciplines such as hydrology and land degradation. | [3] |
| | Geomorphological | Includes both general (detection of simple morphometric features, e.g., peaks, ridges and planes) and specific (detection of discrete features, e.g., sand dunes and alluvial fans) geomorphometry. | [3,6] |
| | Soil mapping | Studies that have employed landforms for digital soil mapping. | [20] |
| | Radiative applications | Use of landforms for computing viewsheds and visibility analysis | [3] |
| RQ3 Analysis Unit | Pixel | Any approach where DEM and/or image cells/pixels are used as input for classification. | |
| | Objects | Any approach where neighboring cells with similar features are segmented into regions, objects, areal features or segments before classification. | [3,10] |
| RQ3 Methods | Deep Learning | Any application of a deep neural network. | |
| | Expert-Knowledge | Any rule-based classification, fuzzy-logic approach or application of pre-existing software tools. | [6,10,21] |

| | | | |
|---|---|---|---|
| | Filter Methods | Any method that employs windows, kernels or neighborhoods to analyze pixels and falls outside traditional expert-knowledge approaches. | |
| | Manual Digitizing | Any classification/identification of landforms that were purely user-driven. | |
| | Supervised | Any classification approach that required user-defined training/labeled data. | |
| | Unsupervised | Any clustering approaches or classifications done using unlabeled data. | |
| RQ4 Validation Techniques | Qualitative | Visual assessment of results. | [22,23] |
| | Quantitative (Pixel-based) | An accuracy metric based on a validation dataset comprising image pixels. | [24,25] |
| | Quantitative (Area-based) | An accuracy metric based on a validation dataset comprising image objects or a comparison based on area measurements. | [25,26] |
| | Quantitative (Pixel & Area-based) | A multi-facet validation approach comprising both Quantitative (Pixel-based) and Quantitative (Area-based) metrics. | |
| | None | No validation of results is reported. | |

## 3. Results and Discussions

### 3.1. Geographical Distribution of Selected Articles (RQ1)

Figure 2 shows the geographical distribution of the selected articles in terms of the study area. The articles selected had a broad spatial coverage, as they were conducted on six of the seven continents (none of the selected articles was conducted in Australia). Many of the studies were located across Europe, with 21 of the 61 (or 34.4%) study sites distributed across the continent (two of the selected articles performed studies across multiple countries, accounting for a total of 61 study areas; additionally, two studies were conducted on a global scale, which were not included in the geographical distribution statistics). Italy had the most studies, with six investigations conducted within its borders. Poland and Greece saw three landform classification studies performed on their terrain, while Portugal had two studies situated across its territory. The remaining European nations recorded only one landform classification study each.

South and Southeast Asia recorded a total of 16 studies (26.2%) performed across the region. China had the most studies out of any country, with ten studies, followed by Iran with six. North America had 11.5% of the selected studies distributed across its region, with most of these studies conducted in the United States. A common trend in many research fields depicts that countries with fewer resources and skilled individuals have a much lower yield in research publications. This trend is also evident in the geographical distribution of the selected articles, where South America and Africa were the least represented regions, with only four and three landform classification studies, respectively. Subsequently, this may indicate that more research regarding landform classification is required across the South American and African continents.

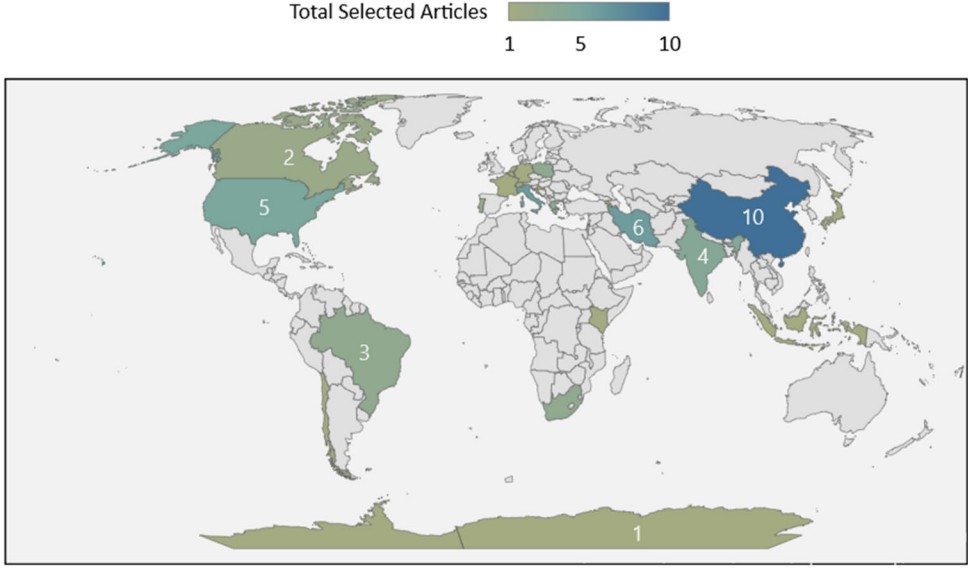

**Figure 2.** Geographical distribution of selected articles shown by country (white numerals show total articles per country).

*3.2. Datasets Used for Classifying Landforms (RQ2).*

The results of the datasets used, sources of DEMs and the cumulated frequency pertaining to the use of different DEMs, including the scales at which they were used, are depicted in Figure 3. Data sources used include very high-resolution DEMs (spatial resolution < 30 m), high-resolution DEMs (spatial resolution = 30 m), moderate to coarse resolution DEMs (spatial resolution > 30 m), digital and laser imagery and analogue and GPS data (Figure 3). Very high-resolution DEMs and high-resolution DEMs are commonly used to delineate landforms at local, regional and national scales, and their frequency of use appears to be comparable (Figure 3). The common use of high-resolution DEMs is most likely due to the recent availability of freely available DEMs datasets covering the entire globe [20,27,28]. Additionally, there appears to be wide use of very high-resolution DEMs available at local scales that are either generated for purposes of the studies or sourced commercially [29–31]. While coarse to moderate resolution DEMs have been used to classify landforms from local to global scales, they were mainly used at regional scales for the studies considered in this scoping review (see Figure 3). Digital imagery, laser-derived DEMs, analogue data and GPS measurements were predominantly incorporated at regional scales. Only one study incorporated a laser-derived DEM at a local scale. The SRTM DEM appears to be the most preferred DEM for landform classification across all scales. While 13 studies used the SRTM DEM at regional scales, it was used once at local, national and global scales for the studies considered in this review. The SRTM DEM's wide use is likely due to it being the first DEM to be freely available at high and moderate resolution at a global level. ASTER GDEM was used in eight studies at regional scales and two at local scales. While TanDEM-X was used in two studies at regional level, USGS-sourced DEM was used in two studies at regional scales and one study at a global scale. ALOS DEM was considered in two studies, one at a local level and the other at a regional scale. This is probably due to ALOS DEM having been launched in 2015, and most researchers appear to readily use the SRTM DEM and ASTER GDEM as they are popular. Commercial and locally sourced DEMs are the most commonly used DEMs for landform classifications at local, regional and national scales (refer to Figure 4). These DEMs were used by 9, 4 and 20 studies at local, national and regional studies, respectively. With regards to the cumulative use of DEMs from 2012 to date, commercial and locally sourced DEMs are the most commonly used followed by the SRTM DEM and the ASTER GDEM. USGS DEMs, TanDEM-X and ALOS DEM did not show signs of an increase in uptake for

landform classification (Figure 5). While the SRTM DEM and locally sourced DEMs were in use since 2012, ASTER GDEM and USGS DEMs were considered from 2013, TanDEM-X from 2014 and the use of ALOS DEM was only observed after 2019.

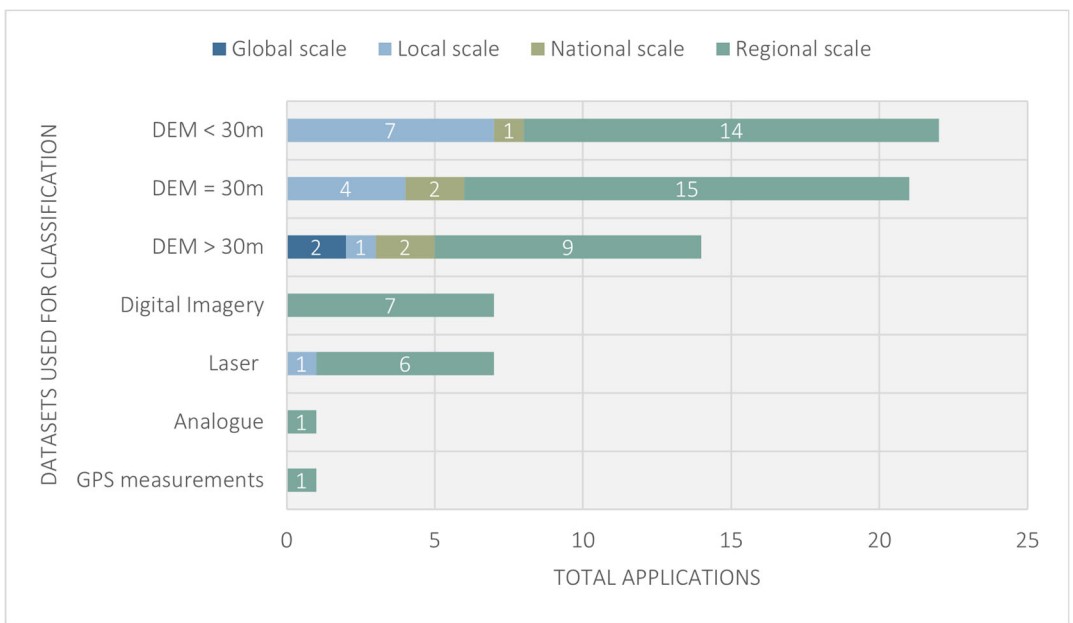

**Figure 3.** Trends of datasets used.

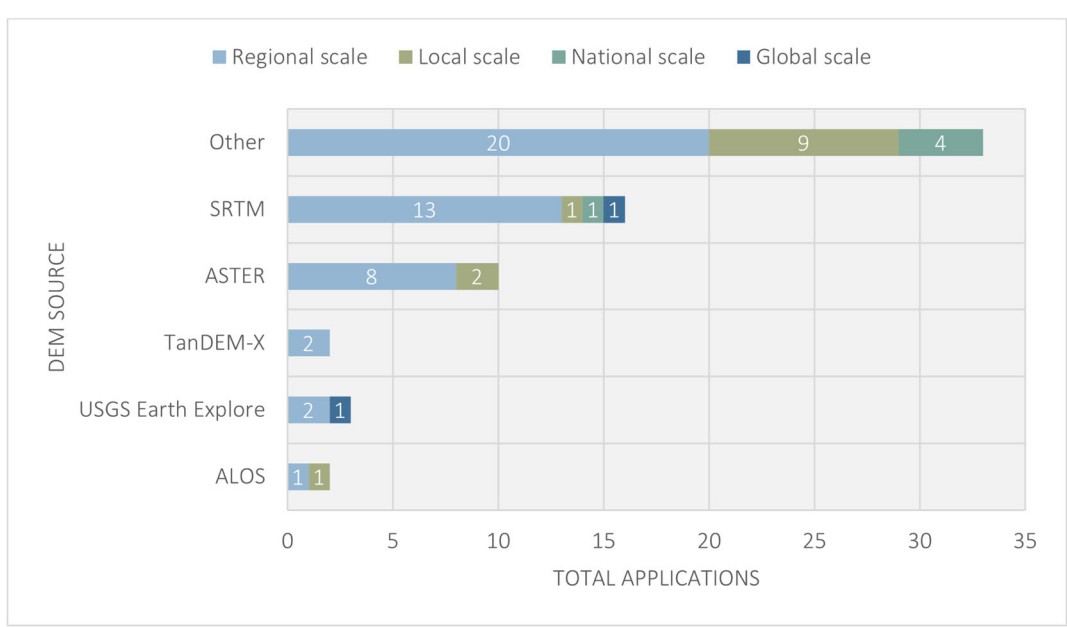

**Figure 4.** DEM sources and scales of applications.

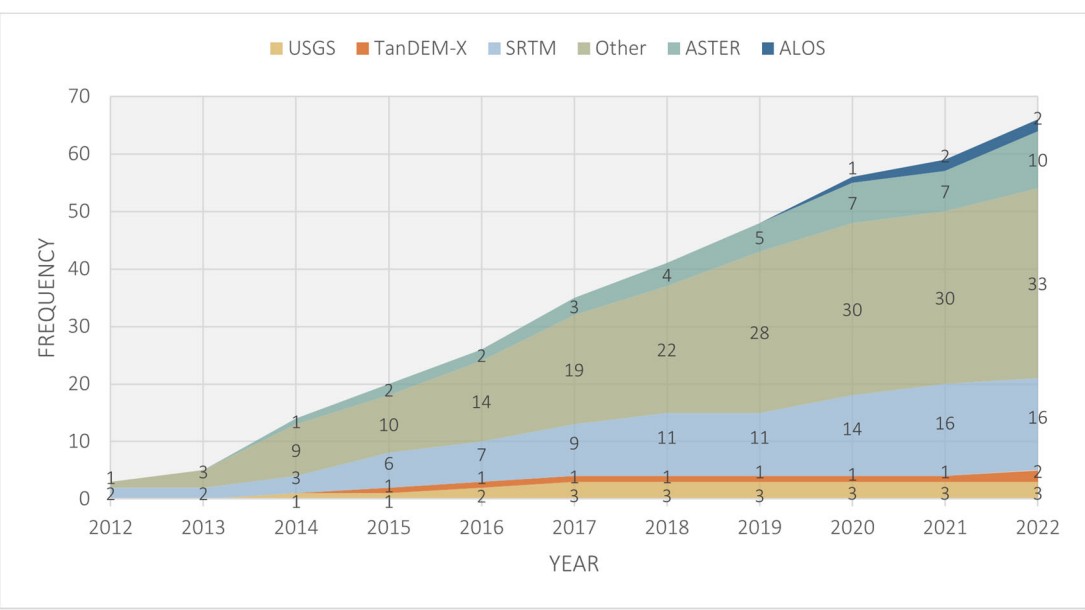

**Figure 5.** Cumulative DEM source.

Overall, it seems that very high-resolution and high-resolution DEMs are the most commonly considered DEMs to delineate landforms at local, regional and national scales. The SRTM DEM appears to be the most used DEM for landform classification at local, regional, national and global levels. The ASTER GDEM has also been considered for landform delineation at local and regional scales. ALOS DEM, USGS DEMs and TanDEM-X have not been widely considered for landform studies thus far.

### 3.3. Analytical Approaches for Classifying Landforms (RQ3).

The results for analysis methods, including cumulative trends, mappings of analysis unit trends, analysis unit versus methods used, analysis unit and scales of application, trends of analysis applications and cumulative trends of applications, are depicted in Figures 6, 7a, 7b, 7c, 8a and 8b, respectively. While most studies were based on expert knowledge, the use of supervised and unsupervised methods was comparable (Figure 6). The dominance of expert-knowledge methods is most probably due to landform classification frameworks being mainly based on predetermined classification schemes. Numerous researchers have, in recent times, simply adopted automated and semi-automated landform classification schemes due to their ease of use and increased availability. Deep learning and filter methods are the least considered methods of landform classification (Figure 6). The less frequent adoption of deep learning approaches is likely due to the fact that machine learning approaches have only become popular recently. The most-used analysis unit is the pixel (Figure 7a). A total of 41 studies used pixel-based methods in comparison to 18 studies that used object-based methods (see Figure 7a). While pixel-based studies used filter methods, deep learning, supervised, unsupervised and expert knowledge, object-based studies were based on unsupervised, supervised and expert knowledge (Figure 7b). Most studies were conducted at regional scales, followed by local, national and global scales (Figure 7c). This is most likely because targeted applications for the respective studies as landforms delineation are scale driven. A total of 2, 5, 11 and 41 studies were conducted at global, national, local and regional levels, respectively. Regarding the scale and the analysis unit of study, most studies conducted at regional level used pixel-based methods (Figure 7c). The number of studies that used pixel-based approaches at a national and local level is comparable. Object-based analysis of landforms appears to be popular at regional and local scales (Figure 7c). Most studies delineated landforms for geomorphological applications, followed by gravity-controlled processes, soil mapping and radiative applications (Figure 8a). As can be seen in Figure 8b, the number of studies

conducted for geomorphological applications has been increasing from 2012 to date. In contrast, not much attention seems to be focused on landform delineation for gravity, radiative and soil mapping applications.

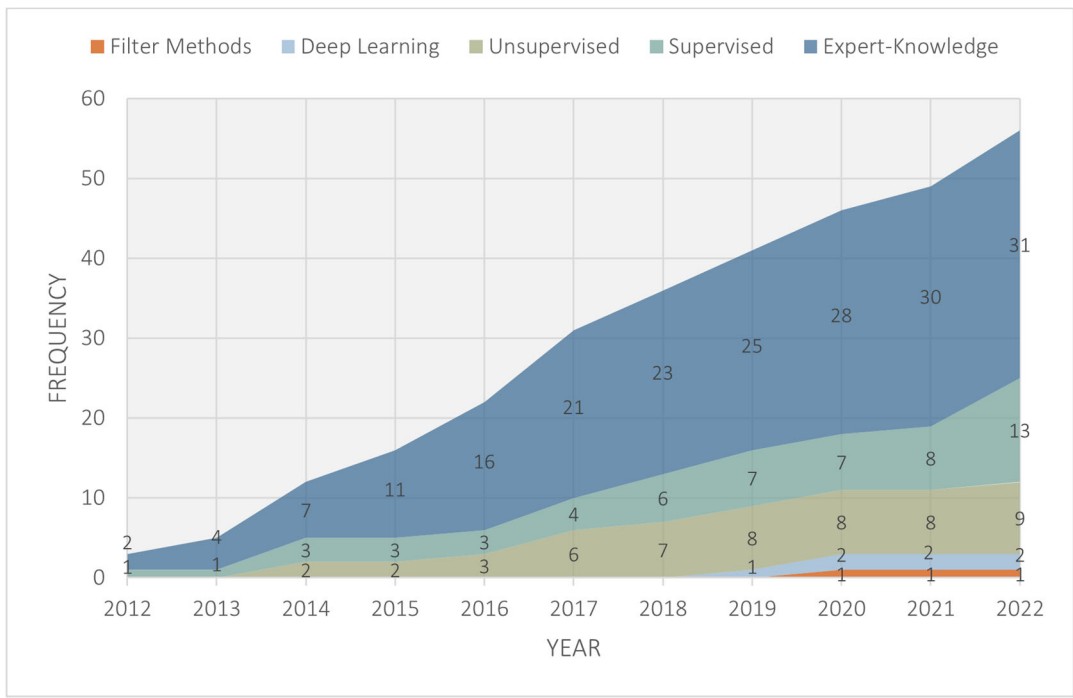

**Figure 6.** Analysis methods and cumulative trends.

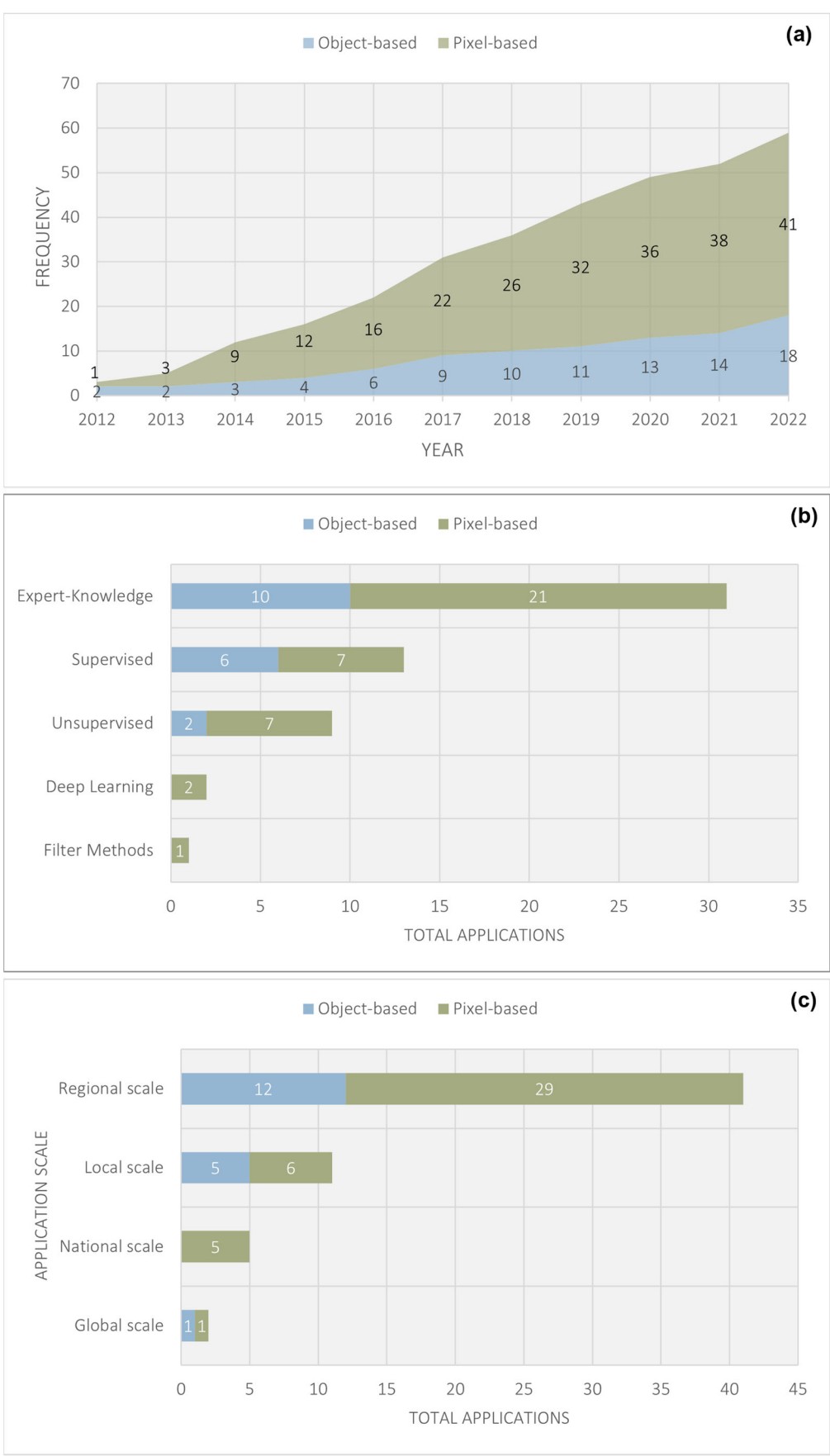

**Figure 7.** Analysis unit cumulative trends (a); analysis unit versus methods (b); analysis unit and scales of application (c).

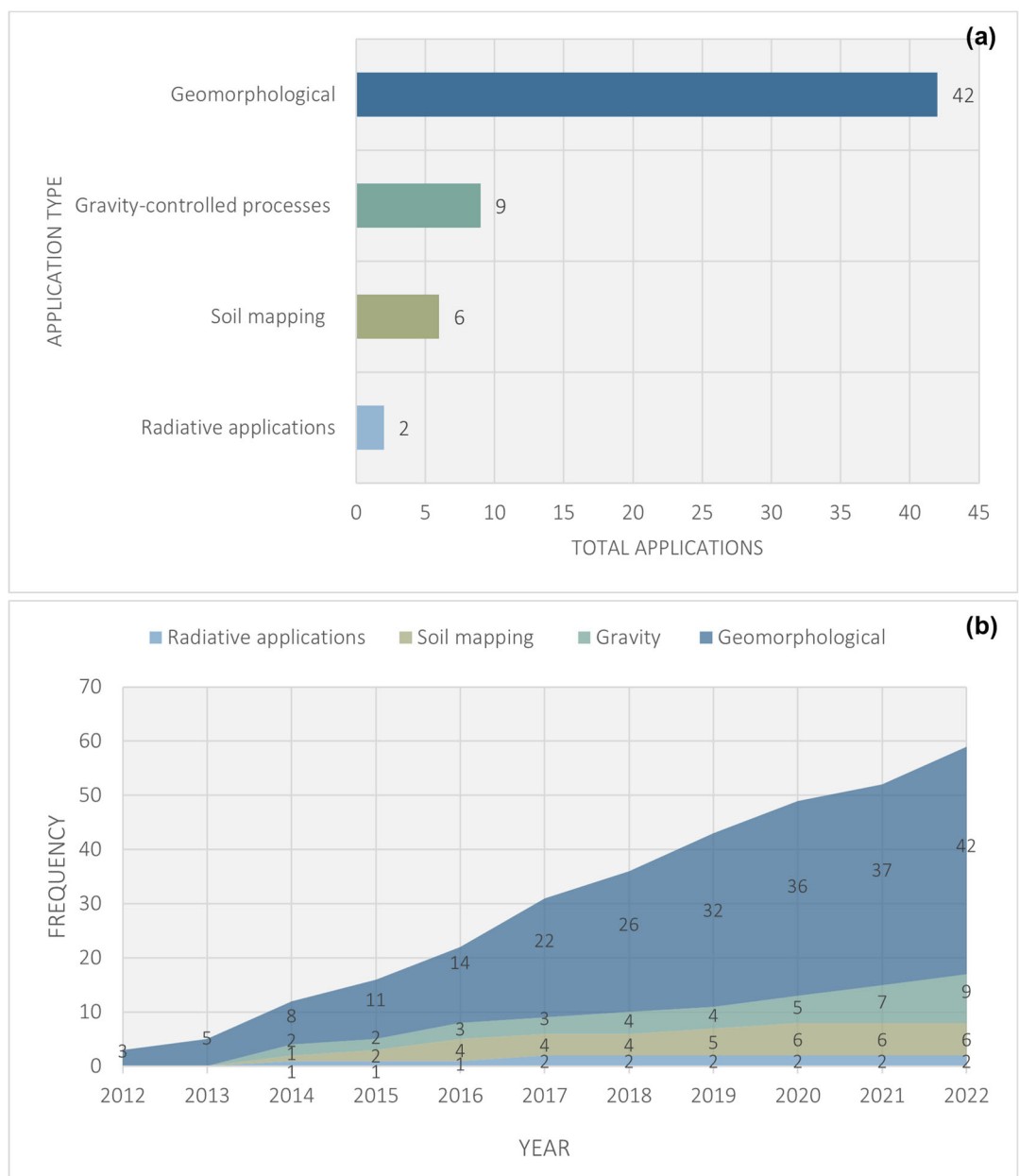

**Figure 8.** Application trends (a); cumulative application trends (b).

Generally, the expert-knowledge approach is the predominant adopted method for landform delineation and has displayed a substantial increase from 2012 to date. Supervised and unsupervised methods are also used. Largely, landform delineation studies use pixel-based approaches. The adoption of object-based methods is increasing steadily though. Geomorphological applications dominate the studies and have increased dramatically in the last ten years, while the other applications have not shown much increase. Deep learning and filter methods seem to have been used after 2019 but have not shown much attraction for landform classification.

### 3.4. Methods Used for Validating Landform Classifications (RQ4).

The validation of landform classification methodologies is essential to assessing the accuracy, robustness, scalability and transferability of the research design. The analysis of the validation processes reported in the selected articles depicted a wide variety of validation methods that have been explored in the literature (Table 3). The analysis indicated

that there was no consensus as to which type of validation is preferred or recommended for a particular application.

**Table 3.** Validation processes reported in the selected articles.

| Validation Process | Total Applications |
| --- | --- |
| Quantitative (Pixel-based) | 17 |
| None | 15 |
| Quantitative (Area-based) | 14 |
| Qualitative | 10 |
| Quantitative (Pixel and Area-based) | 3 |

The trend analysis of the validation methods presented in Figure 9 shows that the majority of the selected articles reported no formal validation procedure between 2014 and 2021. This could be attributed to the increased availability and popularity of expert knowledge-based classification methodologies such as the Topographical Position Index (TPI) [32,33] or the Geomorphons pattern recognition algorithm [34,35]. This notion is corroborated by Figure 10, which illustrates that most studies that employed expert knowledge-based methods reported no validation of the classification results. This trend could suggest that researchers mainly rely on pre-existing classification schemes, most of which have been automated these days.

Additionally, the trend analysis depicts that since 2017, there has been an observable increase in both pixel- and object-based quantitative assessment of classification results. This increase could partly be due to the influx of machine learning applications within the research field, with many machine learning models requiring validation to improve their performance.

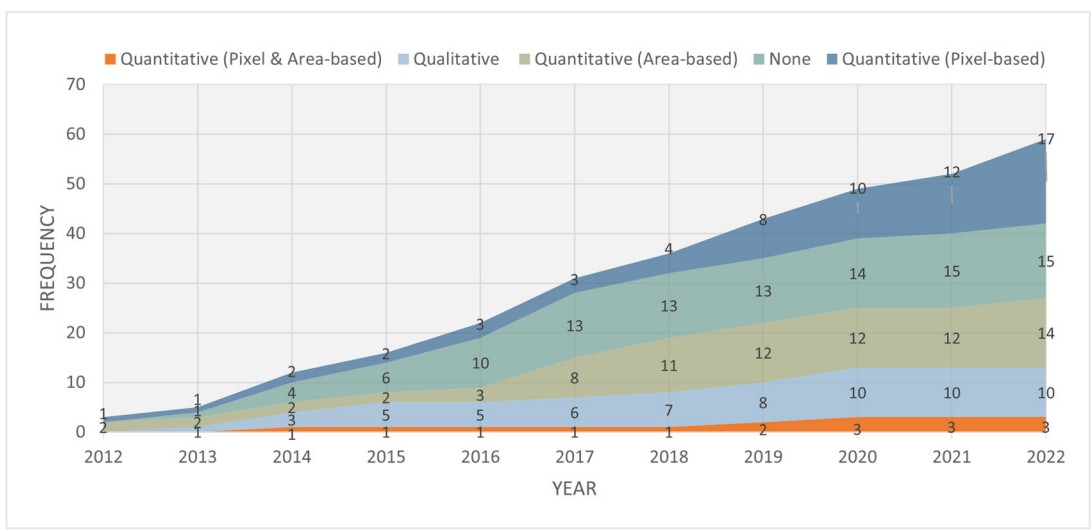

**Figure 9.** Trend analysis of the validation methods reported in the selected articles.

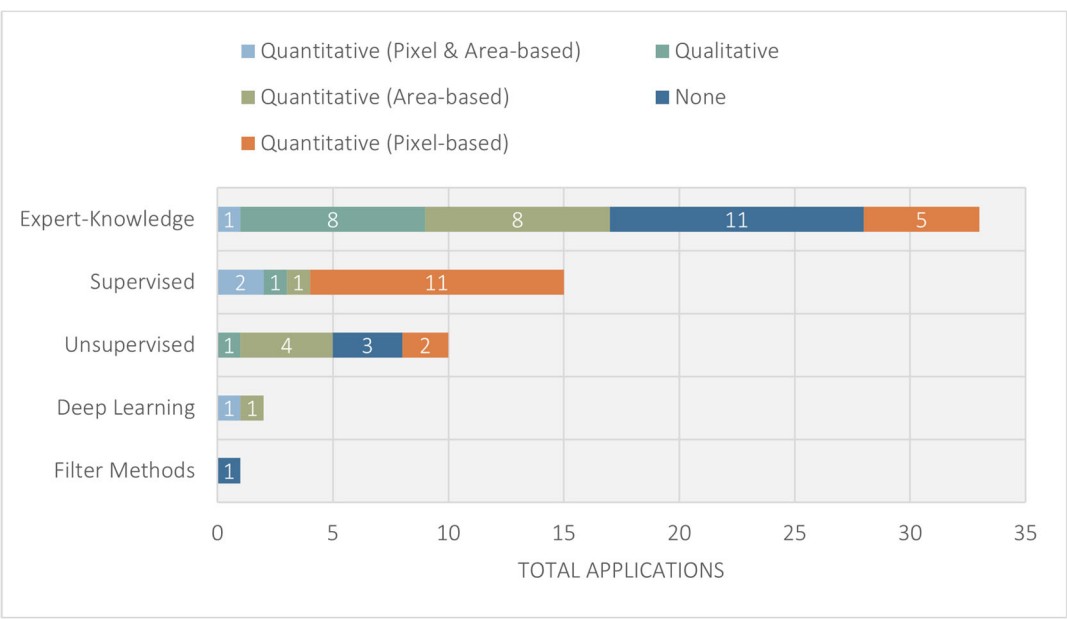

**Figure 10.** Comparison of the validation methods reported in the selected articles and the application types.

Figure 11 exhibits no real trend between the analysis unit (i.e., classification based on pixels or objects) and the validation method. Quantitative (Pixel-based) validation methods are more commonly employed for both pixel- and object-based classification approaches. Although it is common practice to utilize the same analysis unit for the validation process [25], it is evident that Quantitative (Pixel-based) validation methods are still popular among object-based landform classifications. Various authors have argued against using pixels as the validation unit due to their sensitivity to positional errors and its lack of a meaningful relationship with the Earth's features [36,37]. The disconnection of pixels from Earth's features may be problematic for landform delineation as landforms are inherently a spatial area of relatively homogeneous Earth features. Area-based validation processes could be more valuable as they provide an indication of both geometric and thematic accuracy [25]. However, area-based validation processes are more complex compared to pixel-based methods [38]. To date, there still remains uncertainty in the research field regarding what validation processes are optimal for object-based classification [25,38].

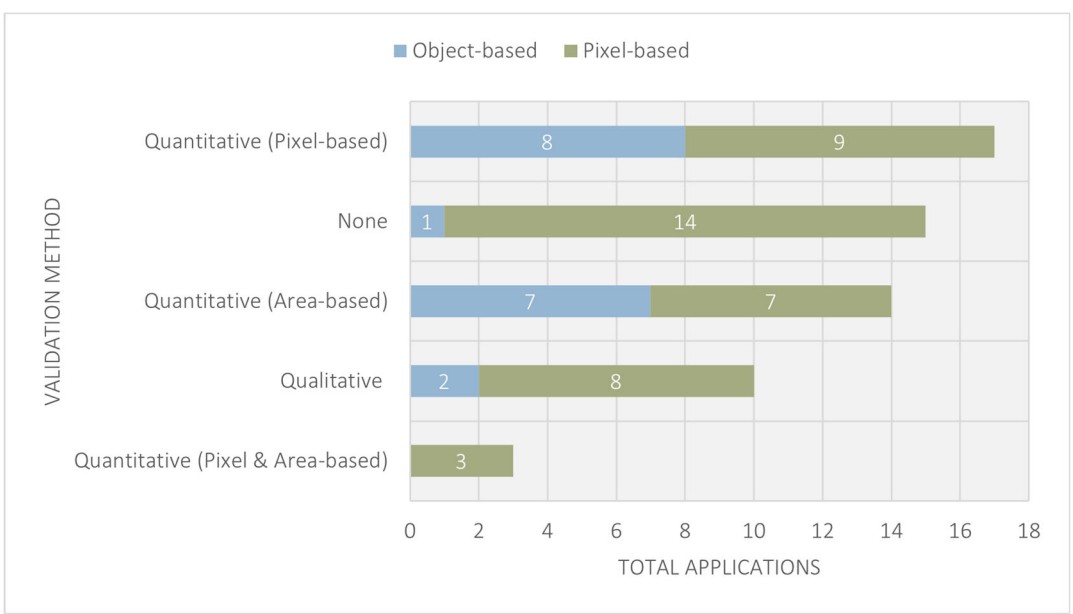

**Figure 11.** Comparison of the validation methods reported in the selected articles and the analysis unit.

Overall, most studies did not report on how landforms were validated. Pixel-based quantitative validation methods are predominantly used for both pixel- and object-based classification approaches. A steady increase in the adoption of quantitative pixel- and object-based validation approached was observed.

## 4. Conclusions

This study explored a scoping review on the use of geospatial methods to delineate landforms. The review set out to map trends in the distribution of studies, datasets used, methods and validation approaches. Four research questions were delineated for this review. They are: where has landform classification been conducted?; what datasets have been used to classify landforms?; how are landforms classified?; and how are landform classification results validated? The resultant articles were systematically screened, of which a total of 59 articles were finally chosen for review, and the relevant parameters were recorded. The parameters were collated and summarized to answer the research questions.

The study found that most landform delineation studies were conducted in Europe, followed by South and East Asia and North America. Very few studies are available that were conducted in South America and the African continents. Regarding the datasets used, the review exposed that the SRTM DEM is the most used freely available global DEM for landform classification at local, regional, national and global levels. This is followed by the ASTER GDEM, which was only used at regional and local scales for studies considered in this review. TanDEM-X, USGS DEMs and ALOS DEM have not been used much for landform delineation. Freely available high-resolution global DEMs and very high-resolution commercial and locally available DEMs products are predominantly considered for landform delineation at local, regional and national scales. Concerning the analysis methods, expert knowledge is the most used approach for landform classification and has been growing in popularity in the last ten years. The use of supervised and unsupervised methods is comparable and has seen steady growth in their use. Filter methods and deep learning are less considered for landform delineation. Most studies are premised on the use of pixels as an analysis unit, but the use of object-based methods is steadily increasing. In comparison to object-based analysis, the adoption of pixel-based analysis for landform delineation has increased substantially in the last ten years. Pixel-based analysis appears to be predominantly more popular than object-based analysis across scales.

With regards to applications, most landform delineation studies were aimed at geomorphological applications. Additionally, there has been a dramatic increase in landform delineation studies for geomorphological applications in the last ten years. Gravity-controlled processes, soil mapping and radiative applications studies were less represented and have not shown much growth in the last ten years. For validation techniques, most studies use quantitative pixel-based methods of validation. A substantial number of studies did not report how landforms were validated. The use of quantitative area-based validation techniques is also common. Qualitative validation approaches are also used fairly often. It appears that the adoption of pixel and object-based quantitative validation techniques are gaining momentum.

The scoping review only considered readily available articles that were published between 2012 and 2022. Only articles that were available in full text and in English were considered. It is likely that crucial studies that did not satisfy the selection criteria were missed. Despite these limitations, we are convinced that this review provides crucial insights into the use of geospatial methods to delineate landforms. We think that there is merit to conduct a full-scale systematic review on this subject to get a clearer picture of the state of research in this field. The SRTM DEM has mainly been used for landform delineation across scales; it would be beneficial to investigate the impact of DEM sources and spatial resolutions on the accuracy of landform classification. This study revealed that the use of very high-resolution commercial and locally sourced DEMs is very common for landform delineation at local, regional and national scales. It would be beneficial to investigate the accuracy of landforms delineated from freely available high-resolution DEM products and very high-resolution commercial and locally sourced DEMs across different scales. While the use of supervised and unsupervised approaches is common for landform classification, the accuracy of landforms delineated through these methods should be explored. In addition, the efficacy of deep learning and filter methods for landform delineation should be investigated. The dominance of landform delineation for geomorphological applications was exposed in this scoping review. Studies that delineate landforms for other applications, for example, soil mapping and radiative applications, should be explored more in future studies. While the use of pixel- and object-based analysis for landform delineation was comparable, the accuracy of these methods for landform delineation requires research attention. In addition, the use of object-based analysis should also be thoroughly investigated. Here, we considered areal objects that were delineated in a variety of ways, but it is not clear how the different methods impact landform delineation. The use of validation techniques also requires research attention. This scoping review uncovered that pixel-based and object-based quantitative techniques are commonly used to validate landforms. A few studies adopted both techniques. It would be beneficial to investigate the value of pixel-based versus object-based quantitative analysis methods for landform validation. Against the backdrop of the popularity of expert-based validation techniques, it is our view that investigations that reveal the accuracy of expert-based versus quantitative approaches should be explored. Seeing that it appears to be common not to validate landform delineation, it is crucial to explore adopting standardized procedures for delineating landforms for different applications.

**Author Contributions:** Conceptualization, Z.E.M.; methodology, Z.E.M. and K.L..; validation, Z.E.M. and K.L.; formal analysis, Z.E.M. and K.L.; investigation, Z.E.M. and K.L.; data curation, Z.E.M. and K.L.; writing—original draft preparation, Z.E.M. and K.L.; writing—review and editing, Z.E.M. and K.L.; visualization, K.L. All authors have read and agreed to the published version of the manuscript.

**Funding:** This research received no external funding.

**Acknowledgments:** We thank Lucinda Raath, Librarian at Stellenbosch University, for assistance to refine the search string and testing on various databases.

**Conflicts of Interest:** The authors declare no conflict of interest.

## Appendix A

**Table A1.** Selected articles.

| Article Index | Title | Authors | Publication year |
|---|---|---|---|
| A1 | GIS-based landform classification of Bronze Age archaeological sites on Crete Island | Argyriou AV, Teeuw RM, Sarris A | 2017 |
| A2 | Multi-resolution soil-landscape characterisation in KwaZulu Natal: Using geomorphons to classify local soilscapes for improved digital geomorphological modelling | Atkinson J, de Clercq W, Rozanov A | 2020 |
| A3 | Using textural analysis for regional landform and landscape mapping, Eastern Guiana Shield | Bugnicourt P, Guitet S, Santos VF, Blanc L, Sotta ED, Barbier N, Couteron P | 2018 |
| A4 | An approach to DEM analysis for landform classification based on local gradients | Camiz S, Poscolieri M | 2018 |
| A5 | Semi-automated object-based landform classification modelling in a part of the Deccan Plateau of central India | Chattaraj S, Srivastava R, Barthwal AK, Giri JD, Mohekar DS, Obi Reddy GP, Daripa A, Chatterji S, Singh SK | 2017 |
| A6 | The land morphology concept and mapping method and its application to mainland Portugal | Cunha NS, Magalhães MR, Domingos T, Abreu MM, Withing K | 2018 |
| A7 | Delineation of main relief subdomains of central Amazonia for regional geomorphometric mapping with SRTM data | de Morisson Valeriano M, de Fátima Rossetti D | 2020 |
| A8 | Application of the topographic position index to heterogeneous landscapes | de Reu J, Bourgeois J, Bats M, Zwertvaegher A, Gelorini V, de Smedt P, Chu Meirvenne M, Verniers J, Crombe P | 2013 |
| A9 | Evaluation of a spatially adaptive approach for land surface classification from digital elevation models | Dekavalla M, Argialas D | 2017 |
| A10 | Automated object-based classification of topography from SRTM data | Drăguţ L, Eisank C, Drăguţ L, Eisank C | 2012 |
| A11 | Multi-modal deep learning for landform recognition | Du L, You X, Li K, Meng L, Cheng G, Xiong L, Wang G | 2019 |
| A12 | Toward geomorphometry of plains - Country-level unsupervised classification of low-relief areas (Poland) | Dyba K, Jasiewicz J | 2022 |
| A13 | Farm-scale soil patterns derived from automated terrain classification | Flynn T, Rozanov A, Ellis F, de Clercq W, Clarke C | 2020 |
| A14 | Detecting and mapping karst landforms using object-based image analysis: Case study: Takht-Soleiman and Parava Mountains, Iran | Garajeh MK, Feizizadeh B, Blaschke T, Lakes T | 2022 |
| A15 | Algorithms vs. surveyors: A comparison of automated landform delineations and surveyed topographic positions from soil mapping in an Alpine environment | Gruber FE, Baruck J, Geitner C | 2017 |
| A16 | Geomorphons — a pattern recognition approach to classification and mapping of landforms | Jasiewicz J, Stepinski TF | 2013 |
| A17 | Modeling global Hammond landform regions from 250-m elevation data | Karagulle D, Frye C, Sayre R, Breyer S, Aniello P, Vaughan R, Wright D | 2017 |
| A18 | Regional topographic classification in the North Shaanxi Loess Plateau based on catchment boundary profiles | Li M, Yang X, Na J, Liu K, Jia Y, Xiong L | 2017 |
| A19 | Employment of Continuous Slope Cumulative Frequency Spectrum in geomorphology quantitative analysis – a case study on Loess Plateau | Lin S, Chen N | 2022 |
| A20 | Automatic Landform Recognition from the Perspective of Watershed Spatial Structure Based on Digital Elevation Models | Lin S, Chen N, He Z | 2022 |

| A21 | Landform classification based on landform geospatial structure – a case study on Loess Plateau of China | Lin S, Xie J, Deng J, Qi M, Chen N | 2022 |
|---|---|---|---|
| A22 | Regional and local topography subdivision and landform mapping using SRTM-derived data; a case study in southeastern Brazil | Manfre LA, de Albuquerque Nobrega RA, Quintanilha JA | 2015 |
| A23 | Classification of landforms in Southern Portugal (Ria Formosa Basin) | Martins FM, Fernandez HM, Isidoro JM, Jordán A, Zavala L | 2016 |
| A24 | Supervised classification of landforms in Arctic mountains | Mithan HT, Hales TC, Cleall PJ | 2019 |
| A25 | Landform classification using topography position index (case study; salt dome of Korsia-Darab Plain, Iran) | Mokarram M, Roshan G, Negahban S | 2015 |
| A26 | Investigation of the relationship between landform classes and electrical conductivity (EC) of water and soil using a fuzzy model in a GIS environment | Mokarram M, Sathyamoorthy D | 2016 |
| A27 | Clustering of landforms using self-organizing maps (SOM) in the west of Fars province | Mokarram M, Sathyamoorthy D | 2016 |
| A28 | Landform classification using a sub-pixel spatial attraction model to increase spatial resolution of digital elevation model (DEM) | Mokarrama M, Hojati M | 2018 |
| A29 | Dynamics of coastal landform features along the southern Tamil Nadu of India by using remote sensing and Geographic Information System | Mujabar PS, Chandrasekar N | 2012 |
| A30 | Object-based large-scale terrain classification combined with segmentation optimization and terrain features: A case study in China | Na J, Ding H, Zhao W, Liu K, Tang G, Pfeifer N | 2021 |
| A31 | Landform pattern recognition and classification for predicting soil types of the Uasin Gishu Plateau, Kenya | Ngunjiri MW, Libohova Z, Owens PR, Schulze DG | 2020 |
| A32 | Predicting Patagonian Landslides: Roles of Forest Cover and Wind Speed | Parra E, Mohr CH, Korup O | 2021 |
| A33 | A GIS procedure for fast topographic characterization of seismic recording stations | Pessina V, Fiorini E | 2014 |
| A34 | Multinomial logistic regression with soil diagnostic features and land surface parameters for soil mapping of Latium (Central Italy) | Piccini C, Marchetti A, Rivieccio R, Napoli R | 2019 |
| A35 | Semi-Automated Classification of Landform Elements in Armenia Based on SRTM DEM using K-Means Unsupervised Classification | Piloyan A, Konečný M | 2017 |
| A36 | Evaluation of TanDEM-X elevation data for geomorphological mapping and interpretation in high mountain environments — A case study from SE Tibet, China | Pipaud I, Loibl D, Lehmkuhl F | 2015 |
| A37 | GIS-based landform and LULC classifications in the Sub-Himalayan Kaljani Basin: Special reference to 2016 Flood | Roy L, Das S | 2021 |
| A38 | Sensitivity of geomorphons to mapping specific landforms from a digital elevation model: A case study of drumlins | Sărășan A, Józsa E, Ardelean AC, Drăguț L | 2019 |
| A39 | Regional-Scale Detection of Fault Scarps and Other Tectonic Landforms: Examples From Northern California | Sare R, Hilley GE, DeLong SB | 2019 |
| A40 | Deep learning-based approach for landform classification from integrated data sources of digital elevation model and imagery | Sijin L, Liyang X, Guoan T, Strobl J | 2020 |
| A41 | Evaluation of machine learning algorithms to classify and map landforms in Antarctica | Siqueira RG, Veloso GV, Fernandes-Filho EI, Francelino MR, Schaefer CE, Corrêa GR | 2022 |

| A42 | LANDFORM ANALYSIS USING TERRAIN ATTRIBUTES. A GIS APPLICATION ON THE ISLAND OF IKARIA (AEGEAN SEA, GREECE). | Skentos A, Ourania A | 2017 |
|-----|------|------|------|
| A43 | Quantifying wetland microtopography with terrestrial laser scanning | Stovall AE, Diamond JS, Slesak RA, McLaughlin DL, Shugart H | 2019 |
| A44 | Linking Geomorphic Process Dominance and the Persistence of Local Elevation | Sweeney KE, Roering JJ, Furbish DJ | 2020 |
| A45 | Comparing data mining classifiers to predict spatial distribution of USDA-family soil groups in Baneh region, Iran | Taghizadeh-Mehrjardi R, Nabiollahi K, Minasny B, Triantafilis J | 2015 |
| A46 | Integrated geomorphological and geospatial analysis for mapping fluvial landforms in Murge Basse Karst of Apulia (southern Italy) | Teofilo G, Gioia D, Spalluto L | 2019 |
| A47 | Biophysical indicators based on spatial hierarchy for informing land reclamation: The case of the Lower Athabasca River (Alberta, Canada) | Thiffault E, Webster K, Lafleur B, Wilson S, Mansuy N | 2017 |
| A48 | Sensitivity of digital elevation models: The scenario from two tropical mountain river basins of the Western Ghats, India | Thomas J, Joseph S, Thrivikramji KP, Arunkumar KS | 2014 |
| A49 | A GIS based method for soil mapping in Sardinia, Italy: A geomatic approach | Vacca A, Loddo S, Melis MT, Funedda A, Puddu R, Verona M, Fanni S, Fantola F, Madrau S, Marrone VA, Serra G, Tore C, Manca D, Pasci S, Puddu MR, Schirru P | 2014 |
| A50 | Using the landform tool to calculate landforms for hydrogeomorphic wetland classification at a country-wide scale | Van Deventer H, Nel J, Maherry A, Mbona N | 2016 |
| A51 | Semi-automated mapping of landforms using multiple point geostatistics | Vannametee E, Babel LV, Hendriks MR, Schuur J, de Jong SM, Bierkens MF, Karssenberg D | 2014 |
| A52 | Object-based landform delineation and classification from DEMs for archaeological predictive mapping | Verhagen P, Drăguţ L | 2012 |
| A53 | Random Forest with semantic tie points for classifying landforms and creating rigorous shaded relief representations | Veronesi F, Hurni L | 2014 |
| A54 | Large-scale spatial variability in loess landforms and their evolution, Luohe River Basin, Chinese Loess Plateau | Wei H, Xiong L, Zhao F, Tang G, Lane SN | 2022 |
| A55 | Identification of topographic elements composition based on landform boundaries from radar interferometry segmentation (preliminary study on digital landform mapping) | Widyatmanti W, Wicaksono I, Syam PD | 2016 |
| A56 | Automatic relief classification versus expert and field based landform classification for the medium-altitude mountain range, the Sudetes, SW Poland | Wieczorek M, Migoń P | 2014 |
| A57 | Soil phosphorus spatial variability due to landform, tillage, and input management: A case study of small watersheds in southwestern Manitoba | Wilson HF, Satchithanantham S, Moulin AP, Glenn AJ | 2016 |
| A58 | Drainage basin object-based method for regional-scale landform classification: a case study of loess area in China | Xiong LY, Zhu AX, Zhang L, Tang GA | 2018 |
| A59 | Landform classification for site evaluation and forest planning: Integration between scientific approach and traditional concept | Zawawi AA, Shiba M, Jemali NJ | 2014 |

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
