# Peer review of "A Scoping Review of Landform Classification Using Geospatial Methods"

_2673-7418, doi:10.3390/geomatics3010005_

Round 1

Reviewer 1 Report

I read the manuscript titled: A scoping review of landform classification using geospatial methods, and I think that the quality of the work is enough good and deserves to be published on Geomatics, following minor revisions.

The abstract shows that the structure of the article is adequate and informative. It describes purpose and contents. Yet, the keywords in some cases are too “wide” and therefore not useful for an effective online search. I suggest to substitute “geomorphology” and “terrain” with more suitable words (or, simply, to eliminate them).

The introduction identifies properly the research problem and it is adequate. The implications of the study are well-developed. Some quotations have to be add.

The methodology outlines the procedures used by the authors for the design of the research, and describes the approach and method of this investigation. I’m not totally in agreement with the authors about the choices of the terms in the search string (section 2.2.), but I understand that the entire works is based on them and therefore those words cannot be changed. In this key, the results are detailed and correspond to the objectives. Further, they are properly presented for an easy understanding.

The discussion is internally consistent and it has a conceptual base background. The authors examine modern and recognized bibliography, but curiously  they not include the papers reported in the Appendix A in the reference list. The conclusions provide new knowledge on the subject matter, but are too short and not exhaustive. Other comments are noted along the manuscript (see the attached file).

Author Response

We thank you for reviewing our manuscript. We are grateful for your comments, they helped improve our manuscript. The responses to the respective comments are attached.

Reviewer 2 Report

Dear authors, I'm very satisfied with your manuscript, without any suggestions or corrections keep in mind its scope, topic and goal. Your contribution is significant with regard to the landforms delineation from different kind of digital elevation models. I just wonder do it is correct to consider 30-m DEMs as the very high resolution ones (today, we have plenty of much detailed 1-m, 5-m, 10-m DEMs covering entire countries)?

Author Response

(The authors gave the same response as above.)

Reviewer 3 Report

The study conducted a literature review of the topic of landform classification using geospatial methods. A systematic and well described procedure was used in the determination of the research questions, collection of studies, and database development. The study selection criteria included keywords, filtering, and research articles published in English between 2012 and 2022. Findings of this study indicated where most studies are conducted, what sources of topographic information, main applications and validation approaches.

General comments:

  • The DEM source “USGS Earth Explorer” needs to be revised. “USGS Earth Explorer” is a website not the source of the DEM. Within the Earth Explorer website, users can download DEMS from multiple sources, including SRTM.

  • Figures 7, 8, and 9 could be combined into a single image with three graphs (a, b, and c).

  • Similarly, Figures 10 and 11 could be combined into a single image with two graphs (a and b).

Itemized comments;

  • L260, “Erath”

  • L269-279, sentence has the word “most” twice.

  • L319. “prowess”

  • L353. “earth’s”

Author Response

(The authors gave the same response as above.)
